# Clinicopathological correlation of kidney disease in HIV infection pre- and post-ART rollout

Nina Elisabeth Diana[1]*, Malcolm Davies[2], Pulane Mosiane[3], Alda Vermeulen[4], Saraladevi Naicker[5]

1 Division of Nephrology, Department of Internal Medicine, Charlotte Maxeke Johannesburg Academic Hospital, Faculty of Health Sciences, University of Witwatersrand, Johannesburg, South Africa, 2 Division of Nephrology, Department of Internal Medicine, Helen Joseph Hospital, Faculty of Health Sciences, University of Witwatersrand, Johannesburg, South Africa, 3 Department of Anatomical Pathology, Charlotte Maxeke Johannesburg Academic Hospital, Faculty of Health Sciences, University of the Witwatersrand, Johannesburg, South Africa, 4 Division of Nephrology, Department of Internal Medicine, Chris Hani Baragwanath Academic Hospital, Faculty of Health Sciences, University of Witwatersrand, Johannesburg, South Africa, 5 Department of Internal Medicine, Faculty of Health Sciences, University of Witwatersrand, Johannesburg, South Africa

☯ These authors contributed equally to this work.
* ninadiana1@gmail.com

**Data Availability Statement:** The final minimized data set underlying this study is available via WIReDSpace, the data repository affiliated with the University of Witwatersrand. https://doi.org/10.54223/uniwitwatersrand-10539-32854.

## Abstract

The spectrum of HIV-associated kidney disease has expanded significantly with the introduction of antiretroviral therapy (ART). In the pre-ART era there was prominence of HIV-associated nephropathy (HIVAN). More recently, the spectrum of disease additionally reflects comorbid illness in the ageing HIV population and ART-related nephrotoxicity. We performed a clinicopathological correlation of kidney disease in HIV-positive individuals who underwent kidney biopsy between 1989 and 2014, utilizing the 2018 Kidney Disease Improving Global Outcomes pathologic classification. ART rollout began in 2004 in South Africa. Patients biopsied pre-ART rollout were compared to those biopsied post-ART rollout with respect to demographics, clinical parameters and histology. We assessed kidney survival in a cohort of these patients following biopsy. Six hundred and ninety biopsies were included, 99 (14.3%) were undertaken pre- and 591 (85.7%) post-ART rollout. Most patients were of Black African descent (97.5%). The post-ART rollout patients were older (p = 0.007), had higher eGFR at presentation (p = 0.016) and fewer presented with eGFR of less than 15ml/min/1.73m$^2$ (p = 0.0008). There was a decrease in the prevalence of classic HIVAN (p = 0.00001); and an increase in FSGS (NOS) in the setting of HIV (p = 0.0022) and tubulointerstitial diseases (p = 0.009) post-ART rollout. Kidney function survival over 5 years was poorest in patients with classic HIVAN (p = 0.00005) and best in minimal change nephropathy (p = 0.0013). Kidney biopsy is crucial for the correct diagnosis and management of HIV-related kidney disease. ART rollout has shifted the spectrum of kidney disease away from classic HIVAN but has not eliminated it. Histological diagnosis prognosticates kidney survival.

**Funding:** The authors received no specific funding for this work.

**Competing interests:** The authors have declared that no competing interests exist.

## Introduction

By the end of 2020, approximately 38 million people were living with HIV, two thirds of whom live in the WHO Africa region [1]. South Africa has the largest HIV epidemic in the world, with 7.8 million HIV-positive individuals. By December 2020, more than 5.6 million people were receiving antiretrovirals in South Africa, the world's largest antiretroviral therapy (ART) program [2].

The kidney is a common target of HIV infection, and an array of acute and chronic kidney syndromes may occur during the course of HIV disease [3]. With the introduction of combination antiretrovirals, a different pattern of kidney disease has been described. In the pre-ART era there was a prominence of HIV associated nephropathy (HIVAN) related to HIV infection of renal cells. More recently, the spectrum of kidney disease reflects comorbid illness in the aging HIV population and ART-related nephrotoxicity [4].

The prevalence of HIV associated chronic kidney disease (CKD) varies geographically from 3.5–48.5% globally [5]. Data on the burden of CKD related to HIV infection in sub-Saharan Africa provides widely varying estimates, depending on the definition used, from 0.7% - 42% [6, 7]. These differences may be accounted for by study design, disparate access to healthcare, scope of ART coverage, AIDS related mortality and genetic diversity.

With the aim of reducing heterogeneity in biopsy definitions of HIV related kidney disease, the KDIGO Controversies Conference on "HIV and Kidney Disease" pathology working group proposed a consensus classification of kidney biopsy findings in the setting of HIV infection [8]. Utilizing this new classification, we have performed a clinicopathological correlation of kidney disease in HIV-positive individuals who underwent a kidney biopsy between January 1989 and December 2014, at two major teaching hospitals affiliated to the University of the Witwatersrand in Johannesburg, South Africa. In addition, we have assessed kidney survival in a cohort of these patients over 5 years following their biopsy.

## Methods

Ethical approval for this study was granted in writing by the Human Research Ethics Committee (Medical) of the University of the Witwatersrand, Johannesburg, South Africa (clearance certificate numbers M1511104, M121184, M120874). This approval permitted a record review of all HIV-positive patients who underwent a kidney biopsy at Charlotte Maxeke Johannesburg Academic Hospital and Chris Hani Baragwanath Academic Hospital within the defined study period. Informed consent for this record review was waived. Data from included patients was anonymised prior to statistical analysis.

Kidney biopsies performed at Charlotte Maxeke Johannesburg Academic Hospital (CMJAH) and Chris Hani Baragwanath Academic Hospital, on HIV-positive individuals, from January 1989 to December 2014 were retrospectively analysed. Demographic data (age, sex and race), clinical parameters (CD4 count, HIV viral load, serum creatinine and urine protein creatinine ratio), indication for biopsy and renal histological pattern was recorded at time of kidney biopsy. The estimated glomerular filtration rate (eGFR) was calculated according to the CKD-EPI creatinine equation without correction for ethnicity. ART rollout began in April 2004 in South Africa. Patients were divided into 2 groups—those who were biopsied pre-ART rollout and those biopsied post-ART rollout. These two groups were compared with respect to the above parameters.

In a subgroup of the patients biopsied between April 2004 and 2014 (post-ART rollout) at CMJAH, additional data laboratory parameters (serum hemoglobin, serum albumin, serial serum creatinine and eGFR) and ART use (at time of biopsy) were recorded.

All kidney biopsies were processed according to standard techniques for light microscopy, immunofluorescence and electron microscopy. All biopsies were reviewed by the National Health Laboratory Service histopathology team who were aware of the HIV status of the patient at time of biopsy.

Histological diagnoses were tabulated using the 2018 Kidney Disease Improving Global Outcomes (KDIGO) Controversies Conference guidelines. As per this guideline, FSGS (NOS) in the setting of HIV describes all non-collapsing forms of FSGS. Those immune complex-mediated glomerulonephritides (ICGNs) with no identifiable comparative etiology other than HIV were categorized as uncharacterized ICGN with no etiology other than HIV. The biopsies with multiple diagnoses were assigned its major clinical-pathological diagnosis for the purposes of analysis.

Shapiro Wilk W testing and visual inspection of the histogram plot indicated non-parametric distribution of baseline characteristics of the cohort; accordingly, central and dispersal measurements were described using the median and interquartile range (IQR), and the Kruskal Wallis ANOVA and Mann-Whitney U tests were used for comparative analyses. Kidney survival, defined by an eGFR above threshold for consideration for dialysis initiation in these institutions ($15mL/min/1.73m^2$), censored for patient default with preserved function, was fitted for 229 patients biopsied post-ART rollout (with follow-up data) using the Kaplan Meyer method; histological diagnoses were compared using Log-rank testing.

## Results

Six hundred and ninety patients living with HIV underwent for-cause native kidney biopsy between 1989 and 2014 (Table 1). The majority of patients were of Black African descent (97.5%); males and females were approximately equally represented (340 males and 350 females). The median age at biopsy was 35 years (IQR 29–41 years).

**Table 1. Characteristics of HIV-positive patients who underwent kidney biopsy.**

| Patient characteristics | Total cohort (N = 690) | Pre-ART rollout (N = 99) | Post-ART rollout (N = 591) | P |
|---|---|---|---|---|
| **Age** (years), median (IQR) | 35 (29–41) | 32 (27–39.5) | 35.5 (30–42) | 0.007 |
| **Sex**, n (%) | | | | |
| Male | 340 (49.3) | 54 (54.5) | 286 (48.4) | 0.257 |
| Female | 350 (50.7) | 45 (45.5) | 305 (51.6) | |
| **Race**, n (%) | | | | |
| Black African | 673 (97.5) | 96 (97) | 577 (97.6) | 0.694 |
| Non-black | 17 (2.5) | 3 (3) | 14 (2.4) | |
| **Laboratory investigations**, median (IQR) | | | | |
| eGFR (ml/min/1.73m$^2$) | 27.1 (11.2–70.1) | 14.9 (7.8–66.8) | 28.5 (12.5–70.1) | 0.016 |
| Patients with eGFR<15ml/min, n (%) | 202 (32.6) | 34 (50.8) | 168 (30.4) | 0.0008 |
| Patients with eGFR>60ml.min, n (%) | 181 (29.2) | 19 (28.4) | 162 (29.3) | 0.873 |
| Urine P:Cr (g/mmol) | 0.52 (0.22–1.01) | 0.33 (0.17–1.14) | 0.52 (0.22–1.01) | 0.488 |
| Patients with Urine P:Cr>0.3g/mmol, n (%) | 313 (64.3) | 11 (55) | 302 (67.0) | 0.268 |
| CD4 count x 10$^6$/L | 222 (99–396) | 174 (57–286) | 229 (100–396) | 0.087 |
| Patients with CD4>500x10$^6$/L, n (%) | 75 (14.0) | 6 (15.8) | 69 (13.7) | 0.740 |
| Patients with CD4<200x10$^6$/L, n (%) | 243 (45.3) | 23 (60.5) | 220 (44.2) | 0.051 |
| HIV VL (RNA copies/ml) | 47884.5 (2592–280795) | 24000 (1350–475000) | 48535 (2592–275490) | 0.190 |

IQR, interquartile range; eGFR, estimated glomerular filtration rate; Urine P:Cr, urine protein creatinine ratio; HIV, human immunodeficiency virus; VL, viral load; RNA, ribonucleic acid

Median eGFR was 27.1ml.min/1.73m$^2$ (IQR 11.7–70.1ml/ min/1.73m$^2$) at the time of biopsy; 32.6% of patients were biopsied at an eGFR of less than 15ml/min/1.73m$^2$. Median proteinuria was 0.52g/mmol (IQR 0.22–1.01g/mmol), and 313 patients (64.3% of the 487 patients with a documented measurement) had nephrotic range proteinuria.

Median CD4 count was 222 x 10$^6$/L (IQR 99–396 x 10$^6$/L). Two hundred and forty-three patients (45.3% of the 536 patients with a recorded measurement) had a CD4 count of less than 200 x 10$^6$/L. HIV viral load was available for 247 patients (9 of these were from the pre-ART rollout group); 73 had viral loads lower than laboratory detectable limit (40 copies/ml). Amongst the remaining 174 patients, median HIV viral load at biopsy was 47884.5 RNA copies/ml (IQR 2592–280795 RNA copies/ml).

Ninety-nine biopsies (14.3% of the total series) were undertaken before the initiation of universal access to antiretroviral therapy (ART) in South Africa (April 2004); 591 (85.7%) were performed after rollout. (Table 1). Patients biopsied after ART rollout group were older (35.5 years compared to 32 years, p = 0.007). This cohort had a higher eGFR at presentation (28.5 ml/min/1.73m$^2$ vs 14.9 ml/min/1.73m$^2$, p = 0.016) and fewer patients with an eGFR of less than 15ml/min/1.73m$^2$ (p = 0.0008). There was no difference in CD4 count and viral load between these two groups; the post-ART rollout group tended towards fewer patients with CD4 counts lower than 200 x 10$^6$/L (p = 0.051).

The most common indications for kidney biopsy in this series were kidney dysfunction (44.9%), followed by nephrotic syndrome (42.9%); isolated proteinuria and albuminuria contributed a further 21 (3%) and 12 (1.7%) cases respectively (Table 2). The nephritic syndrome and isolated hematuria were uncommon indications for biopsy in this series.

Biopsy findings before and after ART rollout in HIV positive patients who underwent a native kidney biopsy are shown in Table 3.

There was a significant reduction in the percentage of glomerular dominant biopsy findings following ART rollout (85.9% to 74.6%, p = 0.015). In the podocytopathies group, there was a significant decrease in the diagnosis of classic HIVAN (43.4% to 22.8%, p = 0.00001) and an increase in FSGS (NOS) in the setting of HIV (4% to 15.6%, p = 0.0022). There was no change in the prevalence of minimal change disease in the setting of HIV or immune complex-mediated glomerular disease on kidney biopsy. Whilst there was no increase in the prevalence of uncharacterised ICGN with no etiology other than HIV (p = 0.718), there was a significant decline in the percentage of cases of endocapillary proliferative and exudative glomerulonephritis in the setting of HIV (8.1% vs 1.2%, p = 0.00001).

Tubulo-interstitial dominant diseases increased from 3% to 11.7% (p = 0.009) following ART rollout, predominantly contributed to by a significant increase in the fraction of patients with tubulointerstitial nephritis (2% to 7.8% p = 0.037). There was no difference in the prevalence of vascular dominant lesions on biopsy.

**Table 2. Indications for biopsy in HIV-positive patients who underwent kidney biopsy.**

| Indication for biopsy, N (%) | Total cohort (N = 690) | Pre ART rollout (N = 99) | Post ART rollout (N = 591) | P |
|---|---|---|---|---|
| Kidney dysfunction | 310 (44.9) | 47 (47.5) | 263 (44.5) | 0.883 |
| Nephrotic syndrome | 296 (42.9) | 40 (40.4) | 256 (43.3) | 0.588 |
| Abnormal urine analysis not otherwise specified | 18 (2.6) | 3 (3) | 15 (2.5) | 0.776 |
| Combined nephritic / nephrotic syndrome | 15 (2.2) | 4 (4) | 11 (1.86) | 0.169 |
| Isolated proteinuria | 21 (3.0) | 0 | 21 (3.6) | 0.057 |
| Albuminuria | 12 (1.7) | 0 | 12 (2) | 0.153 |
| Isolated hematuria | 6 (0.9) | 0 | 6 (1) | 0.314 |
| Nephritic syndrome | 4 (0.6) | 0 | 4 (0.7) | 0.412 |
| Indication not clear retrospectively | 8 (1.2) | 5 (5.1) | 3 (0.5) | 0.0009 |

**Table 3. Biopsy findings before and after ART rollout in HIV-positive patients (N = 690).**

| Biopsy findings, N (%) | Before ART rollout N = 99 | After ART rollout N = 591 | P |
|---|---|---|---|
| *Glomerular-dominant* | 85 (85.9) | 441 (74.6) | 0.015 |
| Podocytopathies | 50 (50.5) | 257 (43.5) | 0.193 |
| Classic HIVAN | 43 (43.4) | 135 (22.8) | 0.00001 |
| FSGS (NOS) in the setting of HIV | 4 (4) | 92 (15.6) | 0.0022 |
| Minimal change disease in the setting of HIV | 3 (3) | 30 (5.1) | 0.377 |
| Immune complex-mediated glomerular disease | 35 (35.4) | 184 (31.1) | 0.404 |
| Uncharacterised ICGN with no etiology other than HIV | 10 (10.1) | 67 (11.3) | 0.718 |
| Membranous nephropathy in the setting of HIV | 8 (8.1) | 37 (6.3) | 0.497 |
| Membranoproliferative glomerulonephritis in the setting of HIV | 6 (6.1) | 38 (6.4) | 0.890 |
| Lupus-like nephritis in the setting of HIV | 1 (1) | 26 (4.3) | 0.108 |
| Endocapillary proliferative and exudative glomerulonephritis in the setting of HIV | 8 (8.1) | 7 (1.2) | 0.00001 |
| IgA nephropathy in the setting of HIV | 1 (1) | 4 (0.7) | 0.717 |
| Immunotactoid glomerulonephritis in the setting of HIV | 0 | 3 (0.5) | 0.477 |
| IgM-dominant immune complex glomerulonephritis in the setting of HIV | 1 (1) | 1 (0.2) | 0.150 |
| Lupus nephritis in the setting of HIV | 0 | 1 (0.2) | 0.857 |
| *Tubulo-interstitial dominant* | 3 (3) | 69 (11.7) | 0.009 |
| Tubulointerstitial nephritis | 2 (2) | 46 (7.8) | 0.037 |
| Acute tubular injury | 1 (1) | 20 (3.4) | 0.203 |
| Pyelonephritis | 0 | 3 (0.5) | 0.477 |
| *Vascular dominant* | 2 (2) | 8 (1.4) | 0.607 |
| Thrombotic microangiopathy in the setting of HIV | 0 | 7 (1.2) | 0.276 |
| Vasculitis not otherwise specified | 2 (2) | 0 | 0.0005 |
| Glomerular ischaemia not otherwise specified | 0 | 1 (0.2) | 0.682 |
| *Other in the setting of HIV infection* | 9 (9.1) | 73 (12.4) | 0.353 |
| Hypertensive nephropathy | 4 (4) | 39 (6.6) | 0.330 |
| Diabetic nephropathy | 4 (4) | 21 (3.6) | 0.810 |
| Pauci-immune vasculitis | 0 | 3 (0.5) | 0.477 |
| Myeloma cast nephropathy | 0 | 2 (0.3) | 0.562 |
| Amyloidosis | 0 | 1 (0.2) | 0.682 |
| C3 glomerulopathy | 0 | 1 (0.2) | 0.682 |
| Advanced chronic injury of uncertain etiology | 1 (1) | 6 (1) | 0.996 |

HIVAN, HIV associated nephropathy; FSGS (NOS), focal segmental glomerulosclerosis (not otherwise specified); ICGN, immune complex-mediated glomerulonephritis

There was no change in the 'Other in the setting of HIV group' (p = 0.353) pre- and post-ART rollout, including the proportion of patients diagnosed with diabetic nephropathy (p = 0.810) and those diagnosed with hypertensive nephropathy (p = 0.33).

Median CD4 count was higher in those patients presenting after ART rollout (229 x $10^6$ cells/L compared to 174 x$10^6$ cells/L, p = 0.087) and increased year-on-year after rollout; the percentage of patients with CD4 count below 200 x $10^6$ cells/L decreased from 60.5% before ART rollout to 44.2% after rollout (p = 0.051). Increasing CD4 count was mirrored by a decreasing annual incidence of classic HIVAN (Fig 1); CD4 count in patients diagnosed with glomerular-dominant pathologies was lowest in those with classic HIVAN (p < 0.0001), (Fig 2). HIV viral load was less frequently suppressed in patients with classic HIVAN and membranoproliferative pattern glomerulonephritis in the setting of HIV (p = 0.613 and p = 0.027, respectively); in contrast, viral load below laboratory detectable limit (viral suppression) was

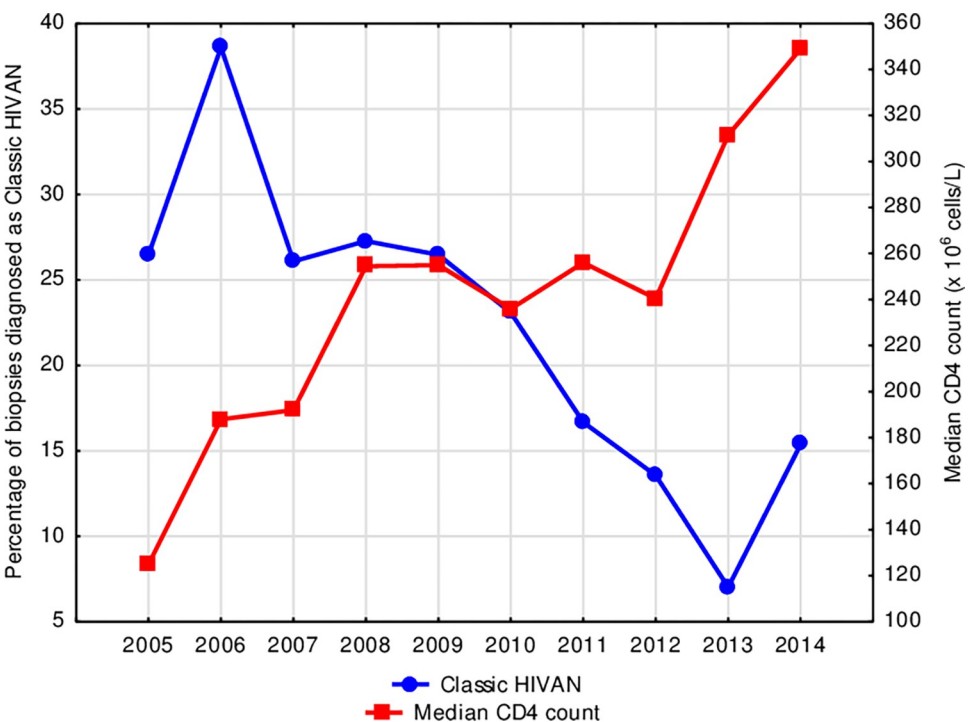

**Fig 1. Median CD4 count and annual incidence of classic HIVAN in HIV-positive patients who underwent a native kidney biopsy following ART rollout in 2004.**

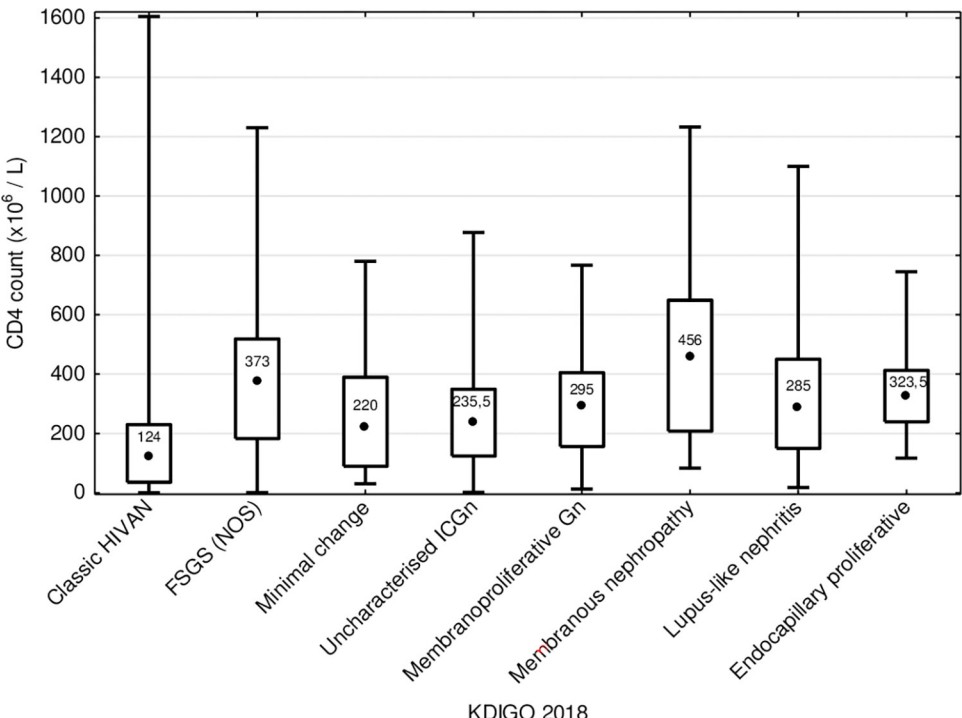

**Fig 2. CD4 count at presentation among glomerular-dominant lesions in HIV-positive patients who underwent a native kidney biopsy.**

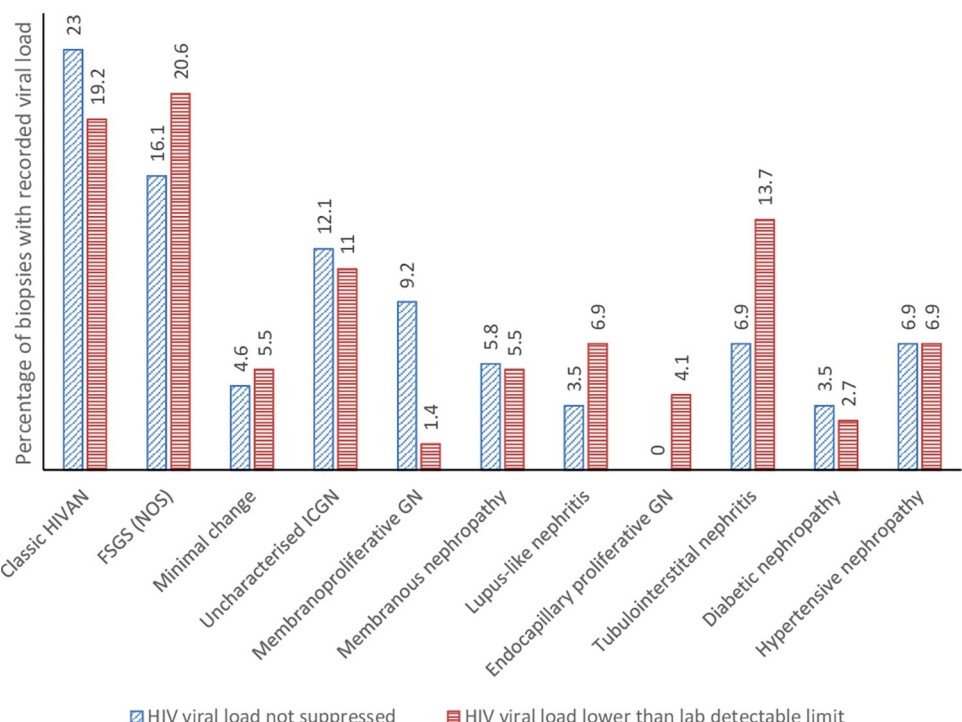

**Fig 3. Percentage of patients achieving viral load suppression among glomerular-dominant lesions in HIV-positive patients who underwent a native kidney biopsy.**

more frequent in FSGS (NOS) and lupus-like nephritis in the setting of HIV (p = 0.462 and p = 0.309, respectively) as well as in biopsies demonstrating tubulointerstitial nephritis (p = 0.326), (Fig 3).

In a subgroup of 229 patients biopsied between April 2004 and 2014, additional data laboratory parameters (serum hemoglobin, serum albumin, serial serum creatinine and eGFR) and ART use (at time of biopsy) were recorded. These patients were followed up for a median duration of 3 months after biopsy (IQR 0–36 months). Baseline and histological characteristics of this group are shown in Tables 4 and 5.

Fig 4 shows eGFR at time of kidney biopsy among patients within the subgroup with a glomerular-dominant lesion; eGFR at time of biopsy was lowest in the classic HIVAN group (21ml/min/1.73m$^2$) compared to other glomerular-dominant lesions (p < 0.0008).

Kidney function survival over 5 years as indicated by an eGFR above 15mL/min/1.73m$^2$ at most recent follow-up was retrospectively available in 229 patients biopsied post-ART rollout; median duration of follow-up in this group was 3 months (IQR 0–36 months). Kidney survival was poorest in those patients diagnosed with classic HIVAN (p = 0.00005) and best in those manifesting minimal change nephropathy (p = 0.0013), Fig 5.

## Discussion

This is the largest kidney biopsy series reported in HIV-positive patients and the largest to assess kidney survival related to histology in this group of patients. It includes 690 patients biopsied at the two main referral teaching hospitals in Johannesburg spanning a 25-year period and evaluates the changes in kidney biopsy findings pre- and post-ART rollout within the same cohort.

**Table 4. Baseline characteristics in 229 HIV positive patients with serial kidney function measurement.**

| Patient characteristics | All (n = 229) |
|---|---|
| **Age** (years), median (IQR) | 35 (30–41) |
| **Sex**, n (%) | |
| Male | 116 (50.7) |
| Female | 113 (49.3) |
| **Race**, n (%) | |
| Black African | 219 (95.6) |
| **Laboratory investigations**, median (IQR) | |
| Hemoglobin (g/dL), (n = 211) | 10 (8.7–11.9) |
| Albumin (g/L), (n = 199) | 27 (19–34) |
| Creatinine (umol/L), (n = 229) | 203 (123–449) |
| eGFR (ml/min/1.73m$^2$), (n = 229) | 34.4 (14.9–68.3) |
| U P:Cr (g/mmol), (n = 202) | 0.51 (0.22–0.95) |
| CD4 count x 10$^6$cell/L (n = 209) | 241 (117–376) |
| HIV VL (RNA copies/ml) (n = 82) | 68000 (1015–302000) |
| **Patients on ART**, n (%), (n = 196) | 81 (41.3) |
| **Patients on ART with HIV VL lower than detectable limit**, n (%), (n = 81) | 41 (50.6) |
| **Duration of follow up** (months) | 3 (0–36) |

IQR, interquartile range; eGFR, estimated glomerular filtration rate; UP:Cr, urine protein creatinine ratio; HIV VL, human immunodeficiency virus viral load

Despite South Africa having the world's largest population of people living with HIV, roll-out of universal access to ART only began at service access points across the country in April 2004 [9]. Charlotte Maxeke Johannesburg Academic Hospital and Chris Hani Baragwanath Academic Hospital were sites included in this initial rollout. Perhaps reflecting the lack of available interventions at the time, only 99 biopsies (14.3%) were undertaken during the period 1989–2004; this number was exceeded in 2005 alone (102 biopsies).

Whilst the age of post-ART rollout period patients is higher than those in the pre-ART roll-out group (p = 0.007), the median age was only 35.5 years (IQR 30–42 years). This reflects patterns of HIV infection in South Africa, adults aged 15–49 years have the highest burden of HIV disease [10]. Improvement of kidney function is usually seen after commencement of ART in patients with HIV associated kidney disease and this was reflected in our cohort in the post-ART rollout group presenting with higher eGFRs (p = 0.016) and fewer patients with eGFR < 15ml/min/1.73m$^2$ (p = 0.0008) [7, 11–13]. Patients with classic HIVAN had lower CD4 counts (p<0.0001) and eGFR (p<0.001) at time of biopsy compared to patients with other glomerular-dominant pathologies. They also had the poorest kidney function survival over 5 years as indicated by an eGFR above 15mL/min/1.73m$^2$. Poorer kidney survival in those patients diagnosed with classic HIVAN is likely to reflect more advanced chronic kidney injury at presentation or possibly the presence of *APOL1* high risk genotypes associated with worse kidney survival [14]. However, *APOL1* status is not available for this cohort.

There were no differences in indications for biopsy between the two groups. The presenting clinical characteristics were also not predictive of biopsy finding, emphasising the importance of kidney biopsy in the diagnosis and management of HIV related kidney disease.

In our cohort, histological diagnoses differed between the pre-ART rollout and the post-ART rollout groups. Glomerular dominant biopsy findings were significantly reduced (p = 0.015). There was no difference in the overall prevalence of podocytopathies (p = 0.193) as the reduction in classic HIVAN was mirrored by an increase in FSGS (NOS) in the setting

**Table 5. Spectrum of biopsy findings in the 229 HIV positive patients with serial kidney function measurement.**

| Biopsy finding | N (%) |
|---|---|
| *Glomerular-dominant* | 175 (76.4) |
| Podocytopathies | 104 (45.4) |
| Classic HIVAN | 49 (21.4) |
| FSGS (NOS) in the setting of HIV | 39 (17.0) |
| Minimal change disease in the setting of HIV | 17 (7.4) |
| Immune complex-mediated glomerular disease | 69 (30.1) |
| Uncharacterised ICGN with no etiology other than HIV | 17 (7.4) |
| Membranous nephropathy in the setting if HIV | 13 (5.7) |
| Membranoproliferative glomerulonephritis in the setting of HIV | 27 (11.8) |
| Lupus-like nephritis in the setting of HIV | 4 (1.7) |
| Endocapillary proliferative and exudative glomerulonephritis in the setting of HIV | 3 (1.3) |
| IgA nephropathy in the setting of HIV | 2 (0.9) |
| Immunotactoid glomerulonephritis in the setting of HIV | 2 (0.9) |
| *Tubulo-interstitial dominant* | 27 (11.81) |
| Tubulointerstitial nephritis | 25 (10.9) |
| Acute tubular injury | 2 (0.9) |
| *Vascular dominant* | 3 (1.3) |
| Thrombotic microangiopathy in the setting of HIV | 3 (1.3) |
| *Other in the setting of HIV infection* | 26 (11.4) |
| Hypertensive nephropathy | 15 (6.6) |
| Diabetic nephropathy | 8 (3.5) |
| Amyloidosis | 1 (0.4) |
| C3 glomerulopathy | 1 (0.4) |
| Advanced chronic injury of uncertain aetiology | 1 (0.4) |

HIVAN, HIV associated nephropathy; FSGS (NOS), focal segmental glomerulosclerosis (not otherwise specified); ICGN, immune complex-mediated glomerulonephritis

of HIV. The significant reduction in the prevalence of classic HIVAN from 43.4% pre-ART rollout to 22.8% post ART rollout (p = 0.00001) is in line with local and international literature which shows a predominance of HIVAN in most biopsy diagnoses prior to availability of ART. In Africa, Han et al, Emem et al and Swanepoel et al, reported biopsy-diagnosed HIVAN in 83%, 70% and 55.3% of their respective cohorts [15–17]. Berliner et al reported on a series of 152 HIV-positive patients who underwent kidney biopsy in the Netherlands, between 1997 and 2004; the diagnosis of HIVAN had decreased from 80% to 20% [18]. Lescure et al. also showed that HIVAN decreased with time between 1995 and 2007 [4]. Kudose et al. reported on a biopsy series between 2011–2018 from a major US centre; HIVAN accounted for only 14% of biopsy diagnoses [19].

Whilst there has been a reduction in the number of classic HIVAN cases seen on biopsy following ART introduction, it has not disappeared completely. In our cohort, this may be related to several factors. Firstly, ART rollout in South Africa was initially slow; by the end of 2010 only 55% of adults eligible for ART in South Africa were receiving treatment [20]. Next is the possibility of non-adherence to treatment in our population. Multiple urban HIV clinics report short term ART adherence rates ranging from 63% to 88% [21–23]. Finally, our cohort was comprised predominantly of Black Africans who are known to have a predisposition to HIVAN due to the prevalence of the *APOL1* high risk genotypes [14].

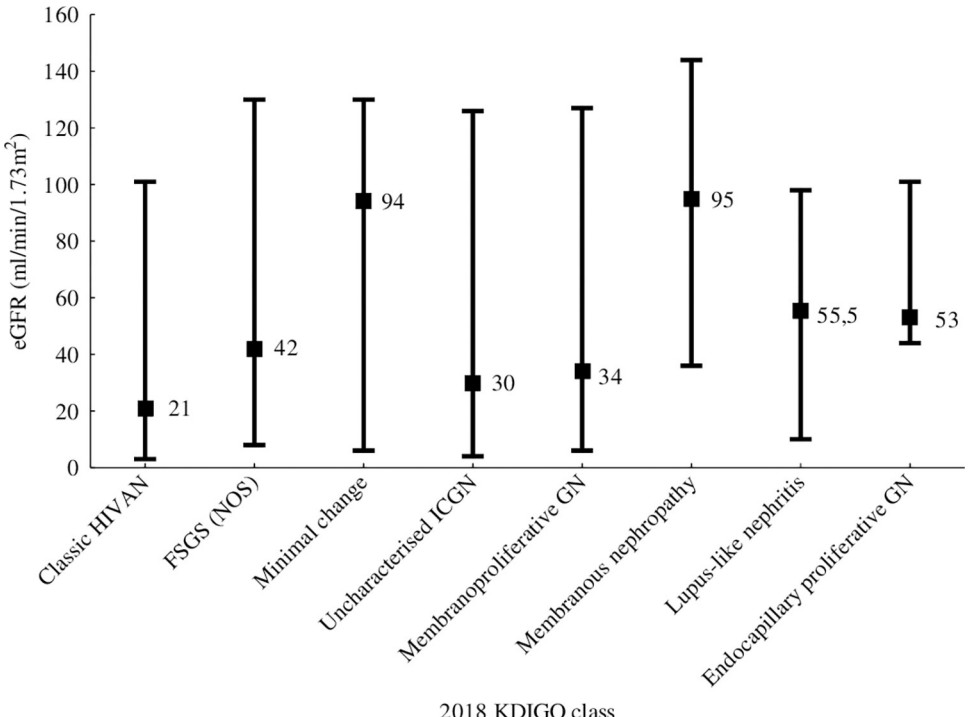

**Fig 4. eGFR at time of kidney biopsy among patients within the subgroup with a glomerular-dominant lesion in the 229 HIV-positive patients with serial kidney function measurement.**

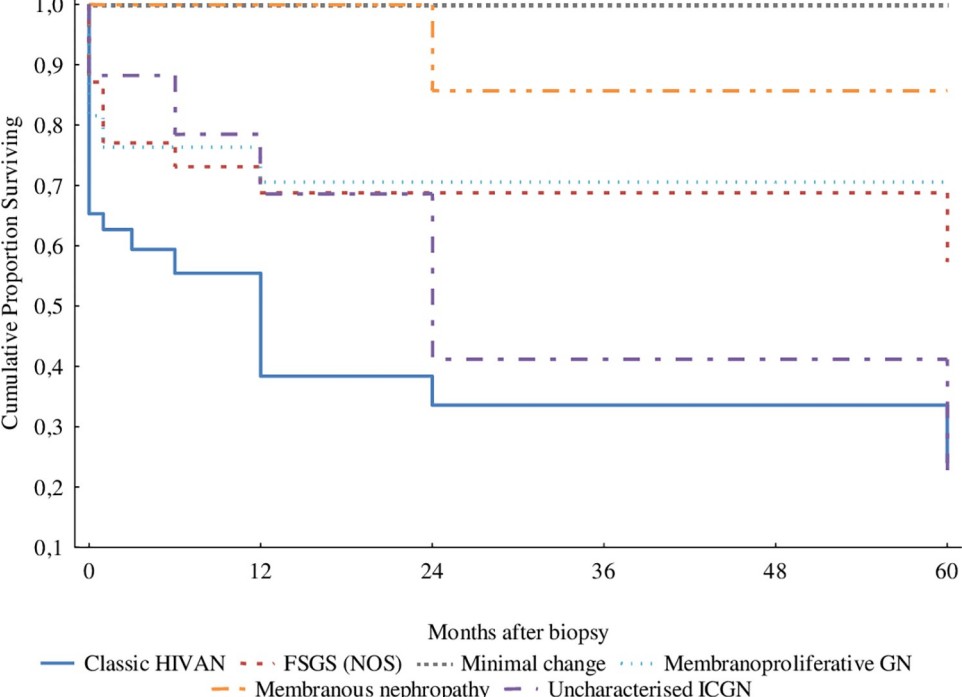

**Fig 5. Kidney function survival by histological type in the 229 HIV-positive patients with serial renal function measurement.**

Our cohort showed an increase in the proportion of histological diagnosis of FSGS (NOS), from 4% pre-ART rollout to 15.6% post rollout (p = 0.0022). FSGS (NOS) in HIV-infected individuals has been hypothesized to represent a partially treated or attenuated form of HIVAN [24]. In our cohort it may be related to the genetic susceptibility to FSGS in individuals of African descent [14]. Both Berliner et al. and Lescure et al. documented an increase in biopsy diagnosis of FSGS with HAART [4, 18]. Kudose et al. reported a 12% prevalence of FSGS (NOS) in the setting of HIV [19].

The percentage of patients with immune complex-mediated glomerular disease in our cohort was similar pre- and post-ART rollout (p = 0.404). The reported prevalence in other African biopsy series is 0% - 40% [12, 16, 25, 26]. The small difference in prevalence of immune complex-medicated glomerular disease between our two groups is in keeping with literature showing incomplete resolution of immune complex deposits following ART [12]. Gerntholz et al. described a 'ball-in-cup' reaction by the basement membrane to sub-epithelial immune deposits [26]. This reaction was seen in a number of biopsies in the uncharacterised ICGN with no etiology other than HIV group. There were few patients from our cohort with IgA nephropathy in the setting of HIV, most likely as IgA nephropathy is less common in patients of black African descent [27]. We did not have access to antiPLA2R antibody testing to assist with identification of patients with primary rather than secondary membranous nephropathy. Data on hepatitis B and hepatitis C status was incomplete, potentially limiting full characterisation of this group.

Tubulointerstitial diseases significantly increased from 3% of all cases pre-ART rollout to 11.7% following ART rollout (p = 0.009). This rise is primarily contributed to by an increase in the number of patients with tubulointerstitial nephritis post-ART rollout (p = 0.037). The use of co-trimoxazole in people living with HIV/ AIDS is considered standard of care [28]. Whilst the World Health Organization (WHO) and the Joint United Nations Programme on HIV/ AIDS (UNAIDS) recommended the use of co-trimoxazole prophylaxis for HIV-positive adults in Africa with symptomatic HIV disease in 2002, it was only in 2006 that the WHO and UNAIDS produced guidelines for national programmes in resource-limited settings [29, 30]. In the absence of clear guidelines, countries and programmes were slow in adopting co-trimoxazole prophylaxis [31]. Co-trimoxazole is a known cause of tubulointerstitial nephritis and its more widespread use following ART rollout may explain the increase in tubulointerstitial nephritis seen in this cohort [32]. Immune reconstitution inflammatory syndrome is an inflammatory disorder associated with paradoxical unmasking or worsening of pre-existing infectious processes after ART initiation which may present as a tubulointerstitial process on biopsy [33]. In 2010 the South African ART guidelines included tenofovir (TDF) as first line therapy [34]. TDF is associated with tubular nephrotoxicity and may account for the increase in acute tubular injury [35]. Kudose et al. reported a prevalence of tubulointerstitial-dominant disease in 26% of their cohort; half were attributed to tenofovir nephrotoxicity [19].

There was no increase in the prevalence of histological diagnoses classified as 'other in the setting of HIV' (p = 0.353) including hypertensive nephropathy (p = 0.330) and diabetic nephropathy (p = 0.810). These are histological diagnoses not consistently linked to HIV infection but more likely to be associated with prolonged life expectancy and traditional CKD risk factors. This is not in keeping with other studies, which report an increase in kidney diseases related to aging and is potentially related to the younger median age of our cohort post-ART rollout (35.5 years [IQR 30–42 years]) [18, 19].

Our study has several limitations. Firstly, this was a retrospective study, so datasets were not complete, particularly CD4 counts and HIV viral loads at time of biopsy and data on use, duration and type of ART was incomplete. Availability of this data would have allowed us to attribute changes in histological patterns to viral suppression. Secondly, the patients' *APOL1*

genotype was not determined in our population known to be high risk. Thirdly, there was a short median follow up duration for our subgroup of patients.

The major strength of this study is its size and span, allowing us to compare patients pre- and post- ART rollout within the same cohort. This size and span in each period may also allow clinical and treatment data flaws due to the retrospective nature of the study to be overcome. Biopsies were performed at the two main referral centres within Johannesburg South Africa and results are generalizable to the population of this region.

In conclusion, kidney biopsy is crucial for correct diagnosis and management of HIV-related kidney disease as it is not possible to predict histology from clinical presentation. Underlying histological diagnosis prognosticates kidney survival but expanded ART use through promotion of patient education and adherence may reduce the burden of HIV-related kidney disease in our resource-limited setting. Newer treatment strategies and prevention of HIV infection, however, are needed to eliminate it.

## Author Contributions

**Conceptualization:** Nina Elisabeth Diana, Malcolm Davies, Saraladevi Naicker.

**Data curation:** Nina Elisabeth Diana, Alda Vermeulen.

**Formal analysis:** Malcolm Davies.

**Investigation:** Nina Elisabeth Diana, Pulane Mosiane, Alda Vermeulen.

**Methodology:** Nina Elisabeth Diana, Malcolm Davies.

**Project administration:** Nina Elisabeth Diana.

**Software:** Malcolm Davies.

**Supervision:** Saraladevi Naicker.

**Validation:** Pulane Mosiane.

**Writing – original draft:** Nina Elisabeth Diana, Malcolm Davies.

**Writing – review & editing:** Nina Elisabeth Diana, Malcolm Davies, Pulane Mosiane, Alda Vermeulen, Saraladevi Naicker.

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
