## [Decision Letter · Decision Letter 0]

23 Mar 2022

PONE-D-22-02569Clinicopathological correlation of kidney disease in HIV infection pre- and post-ART rolloutPLOS ONE

Dear Dr. Diana,

Thank you for submitting your manuscript to PLOS ONE. After careful consideration, we feel that it has merit but does not fully meet PLOS ONE’s publication criteria as it currently stands. Therefore, we invite you to submit a revised version of the manuscript that addresses all the points raised by the two reviewers during the review process.

We look forward to receiving your revised manuscript.

Kind regards,

Emmanuel A. Burdmann

Academic Editor

PLOS ONE

Journal Requirements:

Reviewers' comments:

Reviewer's Responses to Questions

**Comments to the Author**

1. Is the manuscript technically sound, and do the data support the conclusions?

Reviewer #1: Yes

Reviewer #2: Partly

2. Has the statistical analysis been performed appropriately and rigorously? 

Reviewer #1: Yes

Reviewer #2: N/A

3. Have the authors made all data underlying the findings in their manuscript fully available?

Reviewer #1: No

Reviewer #2: No

4. Is the manuscript presented in an intelligible fashion and written in standard English?

Reviewer #1: Yes

Reviewer #2: Yes

5. Review Comments to the Author

Reviewer #1: Diana and colleagues describe HIV kidney diseases before and after the introduction of ART.

General comments. The authors report on 690 kidney biopsies performed on HIV+ patients between 1989 and 2014, likely the largest number in a single report to date. The statistical methods are appropriate and are well-described. The study was approved by an ethics committee.

It is impressive how low eGFR was in both groups at the time of biopsy, 20 and 15. Only 40% were on ART at the time of biopsy after 2004 despite wide-spread availability of ART after 2004. It is striking that fraction with classic HIVAN fell (43% to 23%, p<0.00001) over this time period despite no change in viral loads or CD4 counts.

Specific comments.

1. It is not clear that all patients who were biopsied after ART roll out were receiving and taking ART. Plasma viral load at biopsy would address this issue. If this is not available, this should be acknowledged as a limitation.

2. I am curious as to whether biopsies were performed even with small echogenic kidneys. This is prompted by the fact that 32% had eGFR <15.

3. It would be interesting to hear about biopsy complications. In this context, it be useful to know how the kidneys were visualized (perhaps with ultrasound) and the size (gauge) of the biopsy needles.

4. It is not clear whether Table 3 provides a single diagnosis for each patient or whether a patient could have two diagnoses.

5. All biopsies were read as HIVAN (which I take to mean collapsing glomerulopathy) or FSGS, NOS. This would mean that there were no cases of perihilar variant, tip lesion or cellular variant FSGS. If this is the case, it might be explicitly stated. Alternatively, the term FSGS, NOS may used to mean all non-collapsing variants.

6. Patients were grouped in those whose biopsies were pre-ART (2004) and post-ART (2004). It seems that many post 2004 were not taking ART regularly, as assessed by viral loads, which were similar in the two eras, 19,000 vs 25,000. It would be useful to know the fraction with full viral suppression for each diagnosis in the post-ART era.

Minor comments.

Suggest replacing HIV-infected (off-putting) with HIV-positive, per UNAIDS guidelines:

https://www.unaids.org/sites/default/files/media_asset/2015_terminology_guidelines_en.pdf

P6, consider using sex (biology) instead of gender (social role)

P9. Serum creatinines were “retrospectively available”. I think that you could just say “available.” The reader understands that this is a retrospective study.

In presenting the results, there is some confusion at several places between number of cases and fraction of study group. Thus, on P8:” a significant increase in the number [actually fraction] of patients with tubulointerstitial nephritis, 2% to 7.8%.”

Reviewer #2: The manuscript by Diana N et al brings interesting data regarding renal involvement in HIV patients in two eras, pre and post ART rollout in South Africa. The histological data based on HIV patients’ kidney biopsies encompassed a large period, from 1989 to 2014, including the ART rollout in 2004. Considering the long span in each era, their analysis clearly demonstrated a distinct renal involvement in each period, which has also been demonstrated by other groups. Nevertheless, these data are very interesting because South Africa is the largest HIV epidemic in the world, mostly affecting people of Black African descent, along with the largest ART program. However, the clinical data need some clarification because they are not clearly described in Methods section and some data were only presented as results.

1- Clinical data – when these data (renal function, CD4 count and HIV VL) were collected? What was the actual sample size for each of these parameters? Missing data?

2- Despite the introduction of ART in 2004, how many patients were actually taking the drugs? Authors mention that by 2010, only 55% of the patients were taking drugs. They also mention on line 161, pg. 9, that only 81 pts were taking ART (13% of post ART sample). If this is the case, very few patients were on ART treatment which certainly have little impact on the outcome of the majority of patients included in the study.

3- Renal survival – it is not clear the duration of follow up? Authors mention that serial Cr measurements were available for 232 patients for a median of 3 months (line 174, pg. 9).

Authors should clarify: how many patients were analyzed for renal survival (sample size, pre or post ART rollout, treated or not treated? Duration of follow up?

How they analyzed renal survival in 5 years? Sample size, treated or not, number of patients retained in each period of analysis?

In summary, the histological data pre and post ART rollout seem sound because of the large span in each period which may have overcome clinical and treatment data flaws due to the retrospective nature of the study. Nevertheless, authors should try to improve and clarify the clinical data.

6. PLOS authors have the option to publish the peer review history of their article (what does this mean?). If published, this will include your full peer review and any attached files.

Reviewer #1: **Yes: **Jeffrey B. Kopp

Reviewer #2: No

---

## [Author Response · Author response to Decision Letter 0]

24 Apr 2022

To the Academic Editor and Reviewers

Re: Response letter for re-submission of the manuscript to PLOS One: Clinicopathological correlation of kidney disease in HIV infection pre- and post-ART rollout

Thank you for your review and pertinent comments regarding our manuscript. We have addressed each requirement and provided a point-by-point response to the issues raised during the review process.

Journal Requirements

1. Style requirements

The manuscript has been formatted to meet the PLOS ONE style requirements (including figures and tables). All changed are tracked in the ‘Revised Manuscript with Track Changes’.

2. Ethics statement

The Methods section of the manuscript has been updated to include additional details regarding participant consent.

Ethics clearance was obtained in writing from the Human Research Ethics Committee (Medical) of University of the Witwatersrand, Johannesburg, South Africa; clearance certificate numbers M1511104, M121184, M120874. This is the Ethics Committee affiliated with the academic institution at which the study was conducted. 

This approval permitted a record review of all HIV-positive patients who underwent a kidney biopsy at Charlotte Maxeke Johannesburg Academic Hospital and Chris Hani Baragwanath Academic Hospital within the defined study period. Informed consent was waived due to the retrospective study design. All data was anonymized prior to statistical analysis. 

3. Data Availability statement

The final minimized data set underlying this study is available via WIReDSpace, the data repository affiliated with the University of Witwatersrand. https://doi.org/10.54223/uniwitwatersrand-10539-32854

Response to Reviewer comments

Reviewer 1

1. It’s not clear that all patients who were biopsied after ART rollout were receiving and taking ART. Plasma viral loads would address this issue. If this is not available this should be acknowledged as a limitation.

A sentence noting HIV VL availability and results has been added. HIV VL was available for 247 patients (9 of these were from the pre-ART rollout group). Seventy-three patients had HIV VL lower than detectable limits (Lines 132 - 135, ‘Revised with Track Changes’ manuscript). ART use data was only available for 196 patients (of the 229 patients in the subgroup), 81 (41.3%) of these patients were on ART, of which 41 patients were virally suppressed (data added to table 4, line 205 ‘Revised with Track Changes’ manuscript). This is now noted in the limitations of the study.

2. Whether biopsies were performed even with small echogenic kidneys. Prompted by the fact that 32% had eGFR <15ml/min/1.73m².

Presence of small kidneys is an exclusion criterion for kidney biopsies at both centers where the study was conducted. Biopsy of small kidneys is associated with increased risk of complications and thus was avoided.

3. Biopsy complications and useful to know how the kidneys were visualized and size of biopsy needles.

At both centers kidney biopsies are performed under ultrasound guidance using size 16G automated biopsy needles. Overall complication rates are low but exact data is not available for this cohort.

4. It’s not clear whether Table 3 provides a single diagnosis for each patient or whether the patient could have had two diagnoses.

The Methods section outlines that ‘biopsies with multiple diagnoses were assigned its major clinical-pathological diagnosis for the purposes of analysis’ (Line 104, ‘Revised with Track Changes’ manuscript). 

5. All biopsies were read as HIVAN or FSGS (NOS). This would mean there were no cases of peri-hilar, tip or cellular variant FSGS or alternatively the term FSGS (NOS) may be used to mean all non-collapsing variants.

Histological diagnoses were tabulated using the 2018 Kidney Disease Improving Global Outcomes (KDIGO) Controversies Conference guidelines. These guidelines refer to the term ‘FSGS (NOS) in the setting of HIV’ to describe all non-collapsing forms of FSGS. This has been clarified in the Methods section (Line 101, ‘Revised with Track Changes’ manuscript).

6. Patients were grouped into pre and post ART rollout. Viral loads were similar in the two groups. It would be useful to know the fraction with full viral suppression for each diagnosis in the post-ART era.

Missing data limits the interpretation on viral loads in the post-ART rollout group. In the entire cohort HIV VL was available for 247 patients (only 9 of these were from the pre-ART rollout group); 73 (29.6%) had viral loads lower than laboratory detectable limit (40copies/ml). The patients with HIV VL lower than the detectable limit were excluded from the calculation of median viral loads shown in Table 1. A paragraph (Lines 183 - 188 in the ‘Revised and Track Changes’ manuscript) and Fig 3 (Line 195 in the ‘Revised and Track Changes’ manuscript) have been added to show the percentage of patients achieving viral suppression among glomerular dominant lesions. In the post-ART rollout subgroup of 229 patients, 196 patients had data available regarding ART use, 81 (41.3%) were receiving ART at the time of biopsy and 41 (50.6%) of those patients were virally suppressed.

7. Minor comments

a. Replace HIV-infected with HIV-positive

Changes made to manuscript as suggested

b. Use sex (biology) instead of gender (social role)

Changes made to manuscript as suggested

c. Delete ‘retrospective’ in the sentence on page 9

Retrospective deleted

d. Some confusion between no. of cases and fraction of the study group

The word ‘number’ changed to ‘fraction’ to assist with clarity (Line 170 of ‘Revised with Track Changes’ manuscript).

Reviewer 2

Clinical data needs some clarification because they are not clearly defined in Methods section and only as results

1. Clinical data (renal function, CD4 count and HIV VL) – when were they collected? What is the actual sample size for each of these parameters? Missing data?

The Methods section details the following clinical parameters were collected at time of kidney biopsy for the entire cohort: serum CD4 count, HIV viral load, serum creatinine, eGFR, urine protein creatinine ratio, indication for biopsy and renal histological pattern. The sample size of available data for each parameter has now been noted in the text of Results section (Lines 127 – 135 of ‘Revised with Track Changes’ manuscript). In the subgroup of 229 patients the following additional parameters were collected at time of biopsy: serum hemoglobin, serum albumin and ART use. The sample size of available data for each parameter has now been noted in Table 4 of Results section (Line 205 of ‘Revised with Track Changes’ manuscript). In the subgroup serial creatinine and eGFR measurements were collected (retrospectively) at the following time periods after the biopsy: 3 months, 6 months and then yearly for years one to five.

2. Despite the introduction of ART in 2004, how many patients were actually taking the drugs? Authors mention that by 2010, only 55% of the patients were taking drugs. They also mention on line 161, pg. 9, that only 81 pts were taking ART (13% of post ART sample). If this is the case, very few patients were on ART treatment which certainly have little impact on the outcome of the majority of patients included in the study.

Data relating to ART use was available in the subgroup of 229 patients (all post-ART rollout). In this subgroup, 196 (85.6%) patients had data available regarding ART use, 81 (41.3%) were receiving ART at the time of biopsy. The missing data limits our ability to draw conclusions of ART use in this cohort. Median CD4 count increased year-on-year after rollout; the percentage of patients with CD4 count below 200 x 106 cells/L decreased from 60.5% before ART rollout to 44.2% after rollout (p = 0.051). This may be an indication that there were more than the 81 documented patients on ART in this subgroup. This limitation is now noted in the discussion.

3. Renal survival – it’s not clear the duration of follow up. The following needs clarification: how many patients were analysed for renal survival (sample size, pre or post ART rollout, treated or not treated). How they analysed renal survival in 5 years.

In the post-ART rollout group serial serum creatinine and eGFR measurements were available in the subgroup of 229 patients. Renal survival, defined by an eGFR above 15mL/min/1.73m², censored for patient default with preserved function, was fitted for 229 patients biopsied post-ART rollout (with follow-up data) using the Kaplan Meyer method; histological diagnoses were compared using Log-rank testing (Fig 5, Line 229 of ‘Revised with Track Changes’ manuscript). The Methods section has been updated to make this clearer. Due to this being a retrospective study there were many patients lost to follow up (median duration of follow up was 3 months after biopsy (IQR 0 – 36 months)). 

Thank you for considering our manuscript. We hope to have addressed each requirement appropriately and receive a favorable response to this resubmission.

---

## [Decision Letter · Decision Letter 1]

18 May 2022

Clinicopathological correlation of kidney disease in HIV infection pre- and post-ART rollout

PONE-D-22-02569R1

Dear Dr. Diana,

We’re pleased to inform you that your manuscript has been judged scientifically suitable for publication and will be formally accepted for publication once it meets all outstanding technical requirements.

Kind regards,

Emmanuel A. Burdmann

Section Editor

PLOS ONE

Additional Editor Comments (optional):

Reviewers' comments:

Reviewer's Responses to Questions

**Comments to the Author**

1. If the authors have adequately addressed your comments raised in a previous round of review and you feel that this manuscript is now acceptable for publication, you may indicate that here to bypass the “Comments to the Author” section, enter your conflict of interest statement in the “Confidential to Editor” section, and submit your "Accept" recommendation.

Reviewer #1: All comments have been addressed

Reviewer #2: All comments have been addressed

2. Is the manuscript technically sound, and do the data support the conclusions?

Reviewer #1: Yes

Reviewer #2: Yes

3. Has the statistical analysis been performed appropriately and rigorously? 

Reviewer #1: Yes

Reviewer #2: Yes

4. Have the authors made all data underlying the findings in their manuscript fully available?

Reviewer #1: Yes

Reviewer #2: Yes

5. Is the manuscript presented in an intelligible fashion and written in standard English?

Reviewer #1: Yes

Reviewer #2: Yes

6. Review Comments to the Author

Reviewer #1: The authors have responded well and comprehensively to the reviewers' comments. No new issues have arisen.

Reviewer #2: Authors have addressed most of reviewer's questions despite the significant limitations of their retrospective databank. Nevertheless, authors have extracted the most from the databank and prepared a clear and intelligible manuscript.

There is still one minor point:

The statistical analysis significance should be demonstrated in the figures or in the figure legends. Figures should should be self explained. Readers should not be referred to the text for figure comprehension.

7. PLOS authors have the option to publish the peer review history of their article (what does this mean?). If published, this will include your full peer review and any attached files.

Reviewer #1: **Yes: **Jeffrey B Kopp

Reviewer #2: No

---

## [Editor Report · Acceptance letter]

20 May 2022

PONE-D-22-02569R1 

Clinicopathological correlation of kidney disease in HIV infection pre- and post-ART rollout 

Dear Dr. Diana:

I'm pleased to inform you that your manuscript has been deemed suitable for publication in PLOS ONE. Congratulations! Your manuscript is now with our production department. 

Kind regards, 

on behalf of

Dr. Emmanuel A. Burdmann 

Section Editor

PLOS ONE